# Global Situation of Bioremediation of Leachate-Contaminated Soils by Treatment with Microorganisms: A Systematic Review

**DOI:** 10.3390/microorganisms11040857

**Published:** 2023-03-28

**Authors:** Yesenia Mendoza-Burguete, María de la Luz Pérez-Rea, J. Ledesma-García, Juan Campos-Guillén, M. A. Ramos-López, C. Guzmán, J. A. Rodríguez-Morales

**Affiliations:** Facultad de Ingeniería, Universidad Autónoma de Querétaro, Queretaro 76010, Mexico

**Keywords:** bioremediation, landfill, soil, leachate

## Abstract

This systematic review presents the current state of research in the last five years on contaminants in soils, especially in leachates from solid waste landfills, with emphasis on biological remediation. In this work, the pollutants that can be treated by microorganisms and the results obtained worldwide were studied. All the data obtained were compiled, integrated, and analyzed by soil type, pollutant type, bacterial type, and the countries where these studies were carried out. This review provides reliable data on the contamination of soils worldwide, especially soils contaminated by leachate from municipal landfills. The extent of contamination, treatment objectives, site characteristics, cost, type of microorganisms to be used, and time must be considered when selecting a viable remediation strategy. The results of this study can help develop innovative and applicable methods for evaluating the overall contamination of soil with different contaminants and soil types. These findings can help develop innovative, applicable, and economically feasible methods for the sustainable management of contaminated soils, whether from landfill leachate or other soil types, to reduce or eliminate risk to the environment and human health, and to achieve greater greenery and functionality on the planet.

## 1. Introduction

Inadequate management and disposal of solid waste (SW) is, at best, sent to landfills regulated by environmental standards. However, this is not always the case, because in many developing countries open landfills and dumpsites are widespread, where the majority of all waste generated is disposed of in an unsanitary manner [1]. These open dumps are exposed to a variety of toxins and substances derived from various forms of waste management such as open dumps and open burns, which can cause various health effects in the population ranging from acute health effects such as odor nuisance, headaches, allergies, and skin rashes to more serious and chronic diseases such as respiratory problems, congenital and developmental disorders, and cancer. The link between the increased incidence of disease and exposure to open dumps has been demonstrated by epidemiological and scientific studies [2]. In addition, these open dumps are severely impacted by the pollution of the environment from landfill leachate generated by precipitation, surface runoff, and infiltration water that percolates through the waste. Leachate can move horizontally through the soil, contaminating the soil and damaging vegetation, or it can infiltrate vertically into the soil, often reaching groundwater and aquifers, depending on topographic, geohydrologic, and soil structural conditions. Landfill leachate is the most important secondary pollutant generated by conventional waste treatment. Landfill leachate has high complexity and biotoxicity. It has gradually become a potential threat to environmental safety and human health [3,4].

Remediation technologies include any process that alters the properties of hazardous wastes or contaminants to reduce their toxicity through the application of physiochemical and biological processes [5]. In contrast, biological treatments have attracted much attention in recent years as an effective biotechnological tool for the degradation, redeposition, and transformation of hazardous wastes [6,7]. These biological treatments usually have very little impact on the environment. For example, bioventing, biosparning, bioaugmentation, and microbial bioremediation, which are described in the following section. Project-specific implementation of soil treatment techniques depends on a variety of variables, including the site and its contaminants, bioremediation goals, remediation performance, cost-effectiveness, time, suitability of the public, type of soil, technique, type of microorganisms, sampling methodology, and development of the experiment, which may be laboratory, pilot plant, in situ, or ex situ. Treatability assessments help select the right bioremedial approaches [8]. The results summarized in this systematic review can help develop innovative and applicable methods for assessing global soil pollution. In addition, these results can help develop innovative, applicable, and economical methods for the sustainable management of contaminated soils to mitigate risks to the environment and human health.

### 1.1. Definition of Bioremediation

Bioremediation is a process that uses biological organisms to remove or re-treat an environmental pollutant through metabolic processes and plants to eradicate hazardous pollutants and restore the ecosystem to its original condition [9]. These strategies include natural attenuation, bioaugmentation, or biostimulation, involving microscopic organisms such as fungi, algae, and bacteria. Microorganisms are found in a variety of habitats. They thrive in soil, water, plants, animals, deep water, and ice. Biological mediation technology is widely used and is currently growing exponentially [10].

### 1.2. Bioremediation Techniques

Bioremediation techniques have been the subject of numerous studies and experiments to evaluate their effectiveness and applicability in the removal of certain pollutants. This systematic review focuses mainly on the techniques most suitable for the bioremediation of specific pollutants such as heavy metals, hydrocarbons, polymers and their derivatives, and finally organic pollutants. In the wider applicability of the studies mentioned here, it was found that the use of microbial consortia for the bioremediation of heavy metals [11,12,13,14,15,16] and hydrocarbons [17,18,19,20,21,22,23] gave satisfactory results with significant removals. For the biodegradation of polymers, it was found that certain bacterial strains were able to biodegrade plastics in soil [19,24,25,26,27,28], in contrast to other studies that used bacterial consortia [29,30,31,32], resulting in a significant reduction in these contaminants as well as a reduction in biodegradation time by 90 days [30] and 30 days [31], demonstrating that the symbiotic capacity of bacteria can be a viable option. Finally, there is the bioremediation of organic matter, the most commonly used techniques were the isolation of bacteria [33,34], and some studies specifically investigated the bioremediation capacity of Pseudomonas [35,36,37] and bacterial mixtures [38,39,40,41].

The analysis conducted in this systematic review showed that the most commonly used techniques for bioremediation of soils contaminated by leachate are mainly bioaugmentation, microbiological bioremediation, and phytoremediation, which are discussed in the results section. Each technique has its own advantages and disadvantages, and the choice of a particular technique depends on several factors, such as the nature of the contaminants, the availability of resources, and the degree of soil contamination.

Figure 1, shows several remediation methods, among which phytoremediation [3,42,43,44,45,46,47,48] and microbial bioremediation [43,44,46,49,50] stand out, followed by biostimulation and bioaugmentation [17] among other techniques that are discussed in detail in this systematic review. The importance of the selection of the bioremediation technique, the selection of the microorganisms, the target pollutant, and the experimental scale will be addressed as they directly affect the results obtained. The analysis will focus on empirical studies conducted between January 2017 and January 2023. Although there is no single technique that can remediate different types of soils, there are autochthonous (indigenous) microorganisms that, in symbiosis with inoculated microorganisms, produce results with successful pollutant removal rates. They are the key to solving most problems related to biodegradation and bioremediation of pollutants, provided that environmental conditions are suitable for their growth and metabolism [10,51]. In this study, the importance of the most commonly used biological techniques today is highlighted in order to provide an objective comparison and to promote the development of innovative, applicable, and economically viable methods.

#### 1.2.1. Bioventing technique

The bioventing technique requires controlled stimulation of airflow and small amounts of oxygen to release pollutants into the atmosphere through biodegradation by increasing the activities of indigenous microorganisms. This technique has gained popularity [51,52]. Bioventing is limited by the inability to oxygenate the polluted soil and the inadequate aeration of the surface contamination [10,20,45].

#### 1.2.2. Biosparging Technique

In this technique, air is blown below the water table to raise the oxygen content of the groundwater and accelerate bacterial bioremediation of contaminants [53]. Both techniques, bioventing and biosparging, have been used simultaneously to ensure the efficient removal of soil contaminants despite unfavorable conditions. Biosparging can also combine soil and groundwater to reduce the concentration of dissolved oil compounds in groundwater mixed with soil below the water table and within the capillary fringe. It is a simple and cost-effective method with great flexibility [10].

#### 1.2.3. Bioaugmentation Technique

In bioaugmentation, there are specific sites where microorganisms are needed to extract the pollutants. They are also able to displace indigenous microorganisms, which means they can clean the site quickly. The removal of toxic chemicals through bioaugmentation has already been reported in environments such as soil and water. However, a number of limitations have also been documented [54]. For example, it has been observed that the number of exogenous microorganisms decreases after their addition to a polluted site due to abiotic and biotic stresses. These arise from inadequate growth nutrients such as substrates, temperature fluctuations, and pH, and competition between introduced and indigenous microorganisms [55,56].

#### 1.2.4. Microbial Bioremediation

Bacteria and fungi are the most commonly used microorganisms to remove heavy metals from contaminated soils; although, yeasts and algae are also often used. Bioremediation using microorganisms will be successful when the cultivation of a single strain is replaced by clusters of bacterial strains. The microbes/bacteria used for bioremediation are more than 25 genera that have great potential for Municipal Solid Wastes (MSW) [8,45].

For these reasons, the objective of this review is to provide a background on the involvement of microorganisms in the decontamination of leachate-contaminated soils and their subsequent bioremediation. In addition, the factors limiting the growth of bacteria depending on the environmental conditions are highlighted. Finally, the strategies to improve the decontamination of leachate-contaminated soils based on recently published reports were evaluated through a systematic analysis of the available scientific evidence on the subject over the last five years, highlighting their principles, advantages, limitations, and possible solutions. The prospects for bioremediation are also discussed.

### 1.3. Phytoremediation Technology

Phytoremediation technology is gradually being accepted and used as an effective method for regulating ecological stability and purifying water quality [57]. Phytoremediation techniques purify contaminated soil, water, and groundwater through various pathways, such as phytoextraction, phytofiltration, phytostabilization, phytovolatilization, and phytodegradation (Figure 1) [3,45].

With over 500 million years of evolution, plants have developed highly regulated mechanisms to mitigate toxicity. Some of these heavy metal wastes such as zinc, copper, manganese, nickel, and cobalt are essential trace elements for plant growth [58], or through the unique and selective absorption, transport, and bioaccumulation of plant roots to achieve pollutant degradation and removal. Therefore, phytoremediation has also been frequently combined with other methods.

The efficiency of phytoremediation is often influenced by many factors, such as plant type, pollutant concentration, composition, accumulation, degradation site, auxiliary methods, and even the genotype of the same plant [59].

According to the available research, different plant species have different abilities to remove pollutants, and different genotypes of the same plant have different cleaning efficiency. It was found that broadleaf, composite, sunflower, hemp, and other plants were among the efficient plant metal storage plants, but their cleaning efficiency was closely related to the characteristics of plant species [60].

### 1.4. Theoretical Basis

Obtaining information about platforms is useful. However, keywords are useful for selecting large amounts of information and refining the search methodology to obtain accurate results [61].

Currently, there are different methods and strategies depending on the area in different parts of the world. Common methods include biostimulation, bioaugmentation, biopiles, and bioeradication. All bioremediation techniques have their own advantages and disadvantages, as they have their own specific application [62].

This paper deals with the applications of bioremediation of leachate-contaminated soils treated with microorganisms and the efficiency of each type of strain used, and is organized as follows:

The paper includes a description of different bioremediation techniques with microorganisms and a classification of specific soil types, as well as an analysis of the information and discussions. Finally, the conclusions indicate the contributions of the article and suggestions for future work on bioremediation systems with results according to the microorganisms or strains used, and the most recommended to obtain better treatment results in soils contaminated with landfill leachate.

## 2. Materials and Methods

This paper presents a comprehensive systematic review of the published scientific literature on bioremediation of leachate-contaminated soils in terms of techniques used and percent removal of contaminants potentially harmful to living organisms. PRISMA guidelines were followed at various stages in the preparation.

### 2.1. Initial Search

For this systematic review, a search was conducted for English-language journal articles published between January 2017 to January 2023 in the Scopus, Science Direct, and PubMed databases and in Google Scholar using combinations of the Boolean operators AND, OR, and NOT, as appropriate. These searches yielded a substantial number of results, many of which were repetitive or not very useful to the review, but they provided an overview of the breadth of the topic and allowed us to verify that only one non-systematic review had previously been conducted on this topic [63].

### 2.2. Selection of Articles

Various combinations of terms were used for this search using Boolean operators for each of the selected search platforms, using the combination that yielded the best results in the advanced search engines, as shown in Table 1.

A total of 703 results were obtained. Before starting the selection of articles, the inclusion and exclusion criteria were defined.

#### 2.2.1. Inclusion Criteria of the Search

They must be empirical research and reviews, not single case studies, books, or manuals.They must use experimental techniques with scientifically sound results.The articles or reviews must be studied in a biological context.Results and conclusions must be clear to avoid confusion in synthesis and interpretation of objectives.Published between 2017 and 2023.

#### 2.2.2. Exclusion Criteria

Studies related to the bioremediation of hydrocarbon soils.Studies that address remediation of soils without the use of biotechnology.Research that addresses bioremediation in contexts other than bioremediation of leachate-contaminated soils, landfills, or other approaches that are not within the scope of this systematic review.

### 2.3. Systematical Search

Using these criteria, duplicate articles were first checked so that 138 articles could be excluded. Then, the titles of each article were reviewed, and 219 articles were excluded because they were unrelated or had no relevant contribution to this study. For the eligibility review, each abstract was read, resulting in 152 articles that were included in the analysis of this systematic review. Figure 2 shows a summary of the selection stages of the articles studied.

The bioremediation of soils contaminated with leachate is a subject that has not yet been studied in depth; although, there are scientific publications that demonstrate the presence of various pollutants that are potentially hazardous to water, soil, and air, and therefore harmful to living organisms [35].

Figure 3 shows the articles reviewed and considered in the last 5 years according to the PubMed, Web of Science, Scopus, and Google Scholar platforms. The figure shows the increase in research papers in the last 2 years, which is due to the fact that in the last decades, the need for remediation of contaminated soils has arisen.

### 2.4. Data Classification

Parameters were extracted from each of the articles included for this systematic review: technique, microorganism, contaminant, methodology, and percent removal of contaminant.

The search results of the 152 articles are discussed in detail below, beginning with studies on bioremediation of leachate-contaminated soils in landfills, the countries that have contributed to research on bioremediation of leachate-contaminated soils, and the predominant techniques and types of microorganisms most commonly used.

## 3. Results and Discussion

Of the 152 articles selected for this systematic review, only 21 studies deal 100% with soils contaminated with open dump leachate, showing different types of contaminants affecting the environment. These were classified according to the type of pollutant targeted for bioremediation. Therefore, it is important to understand the definition of incubation and inoculation and the isolation of bacteria. Incubation and inoculation are terms commonly used in the field of microbiology and biotechnology. Incubation refers to the process of creating optimal conditions for the growth and multiplication of microorganisms. This may include the regulation of temperature, humidity, pH, and nutrients in the culture medium. Incubation is commonly used in microbiological studies to grow microorganisms so that they can be analyzed and characterized [26,64].

Inoculation is the addition of microorganisms to a culture medium or substrate to allow their growth and multiplication. In the scientific literature, inoculation is used in a variety of contexts, including biomass biofuel production, environmental remediation, and fermented food production. The inoculation process is a central component of biotechnology and applied microbiology [65,66]. On the other hand, bacterial isolation in soil bioremediation refers to the separation of specific microorganisms for use in the degradation of contaminants in soil [66]. The technique involves the isolation and identification of specific microorganisms such as bacteria and fungi that are capable of degrading contaminants in soil [67].

Bioremediation is a sustainable and effective approach to remediate contaminated soils, and isolation of bacteria is an important technique to improve the effectiveness of bioremediation. This technique selects and cultivates specific strains of microorganisms capable of degrading specific contaminants. Some studies have shown that isolation of bacteria improves the effectiveness and efficiency of bioremediation. For example, the study by Hassan et al. (2019) presented in Table 2 shows that bacterial isolation of specific microorganisms improves the efficiency of bioremediation of soils contaminated with heavy metals. Table 3 lists the bacteria and techniques used for polymers, plastics, and phthalates. The latter are used as additives in plastic manufacturing and have been able to leach into the soil, posing enormous risks to the environment and human health. Plastics, in turn, are environmental pollutants that are produced in large quantities [68]. Fortunately, there is sufficient scientific evidence for the existence of bacteria that degrade these types of pollutants. For example, the study by Kumar et al. (2021) shows that the isolation of bacterial consortia is successful for the degradation of plastics such as polyethylene (PE), polyethylene terephthalate (PET), and polystyrene (PS), which suggests that the isolation of bacteria is a potential technique. It is known that there are a variety of hydrocarbons, such as aromatic hydrocarbons, whose physical and chemical properties can vary depending on the source of the reservoir. Hydrocarbons or organic compounds are extremely insoluble in water.

Microorganisms can oxidize or ferment hydrocarbons, depending on whether these metabolic pathways are present. Industrial activities, petroleum and its derivatives, and incomplete combustion of fossil fuels cause environmental pollution. In fact, petroleum and its derivatives have very serious environmental impacts on contaminated marine and terrestrial habitats [22,69]. For example, the study by Swati et al. (2020) presented in Table 4 shows that the technique of isolation and inoculation of specific bacteria such as *Pseudomonas* sp. can remove 97% of pynene, and in Table 5, Liu et al. (2020) show that 20 to 40% of contaminants, such as particulate organic matter, can be removed (POM).

**Table 2 microorganisms-11-00857-t002:** The removal rate of Heavy Metals.

Year	Microorganism Used	Scale	Contamination in Soil	Method Used	Results/Percentage Removal of Contaminant	Reference
2017	Gram-positive	Laboratory	metals	Isolation and Inoculation	The microbes demonstrated the ability to tolerate and resist metals at different concentrations	[70]
2017	*Bacillus* sp., *Lysinibacillus* sp., and *Rhodococcus* sp.	Laboratory	heavy metal	Bioaugmentation	Remediation percentage between 41% and 88%	[71]
2017	*Bacillus* sp., *Lysinibacillus* sp. and *Rodococcus* sp.	Laboratory	heavy metal	Isolation and inoculation	Difficult bioremediation for the individual microbes, efficiency (above 50%) except when blended in the appropriate formulation.	[72]
2019	*Eisenia fetida Sav*	In Situ	Heavy metal	Zooremediation	High resistance to contamination, which was confirmed by the study.	[73]
2019	*Perenniporia subtephropora*, *Daldinia starbaeckii*, *Phanerochaete concrescens*, and others	Laboratory	Heavy metal	Isolation and Inoculation	Remediation percentage between 38% and 62%	[74]
2020	*Bacillus* sp.	Laboratory	Cr (VI)	Isolation and Inoculation	Effective remediation of toxic Cr (VI)	[75]
2020	*Ascomycota*, *Basidiomycota*	Laboratory	metals and metalloid	Bioaugmentation	Remediation percentage between 52% and 77%	[76]
2020	*Basidiomycota*, *Ascomycota*	Laboratory	Heavy metal	Isolation and Inoculation	Remediation percentage between 36% and 52%	[77]
2022	*Festuca arundinacea*	In Situ	Pb	Phytoremediation	Remediation percentage between 83% and 89%	[78]

**Table 3 microorganisms-11-00857-t003:** The removal rate of polymers and phthalate.

Year	Microorganism Used	Scale	Contamination in Soil	Method Used	Results/Percentage Removal of Contaminant	Reference
2019	*Pseudomonas* sp.	Laboratory	Diethyl Phthalate (DEP)	Isolation and Inoculation	Degradation capability 500 mgL^−1^ in 12 h	[29]
2019	*Paenibacillus* sp.	Laboratory	Low-density polyethylene (LDPE)	Isolation and Inoculation	This isolate could be used for bioremediation as a promising tool for polyethylene degradation.	[79]
2021	*Proteobacteria*, *Bacteroidetes*, *Firmicutes*, and *Actinobacteria*	Laboratory	Polyethylene (PE), polyethylene terephthalate (PET), and polystyrene (PS).	Isolation and Inoculation	Further studies are required for reconstruction of the genome to determine potential role in plastic waste degradation	[25]
2022	*Paenarthrobacter* sp.	Laboratory	Phthalic acid esters	Incubation	Degradation capability 1000 mgL^−1^ in 15 h	[80]

**Table 4 microorganisms-11-00857-t004:** The removal rate of Hydrocarbons.

Year	Microorganism Used	Scale	Contamination in Soil	Method Used	Results/Percentage Removal of Contaminant	Reference
2019	*Proteobacteria*, *Bacteroidetes*, *Firmicutes*, *Cyanobacteria*, and others	Laboratory	Polycyclic aromatic hydrocarbons	Natural attenuation and biostimulation	Up to 66.6%	[19]
2020	*Pseudomonas* sp.	Laboratory	Pyrene	Isolation and Inoculation	Degrade upto 97% of pyrene	[81]
2022	*Daphnia magna* and *Allivibrio fischeri*	Laboratory	Petroleum hydrocarbons (TPH), poly-aromatic hydrocarbons (PAHs)	pyrolysis method	Remediation percentage 99%	[82]

**Table 5 microorganisms-11-00857-t005:** The removal rate of Organic pollutants, atmospheric pollutants, and others.

Year	Microorganism Used	Scale	Contamination in Soil	Method Used	Results/Percentage Removal of Contaminant	Reference
2020	*Firmicutes* and *Proteobacteria*	Laboratory	Particulate organic matter (POM)	Isolation and Inoculation	Remediation percentage between 20% and 40%	[37]
2020	Bacterial community. Not specified	Laboratory/In Situ	H_2_S, NH_3_, and VOCs.	biochar	Remediation percentage between 95.43% and 100%	[83]
2020	*Typha latifolia*	Laboratory	Na(+) and Cl(−)	Isolation and Inoculation	high efficiency of hydro-ponic system for nutrient and salinity removal.	[84]
2020	*Autochthonous fungi*, *Aspergillus flavus*, *Aspergillus niger*, *Fusarium solani*, and others	Laboratory	Organic pollutants	Isolation and Inoculation	Remediation percentage between 22% and 34%	[39]
2022	Gram-positive and Gram-negative	Laboratory	toxic elements in the environment	Isolation and Inoculation	Remediation percentage: 33.35%	[85]

In summary, the isolation and inoculation of bacteria in soil bioremediation is an important technique to improve the effectiveness and efficiency of bioremediation. This technique involves the isolation and identification of specific microorganisms capable of degrading certain contaminants in soil. The study by Qin et al. (2020) shows that pollutants such as H_2_S, NH_3_, and volatile organic carbon can be removed even 100%, which depends on the technique and bacteria used.

### 3.1. Contamination of Leachate-Contaminated Soils Worldwide

As shown in Table 2, Table 3, Table 4 and Table 5 of the previous section, landfill leachate contains a variety of pollutants, most of which are harmful to human health and the environment. Some of the major pollutants found in landfill leachate are:

**Heavy metals:** such as lead, mercury, cadmium, and arsenic, which, in some cases, cause neurological damage, liver damage, cancer, and other health problems [47,86,87].

**Toxic organic compounds:** such as benzene, toluene, chloroform, and other solvents, which cause central nervous system damage, respiratory diseases, and other health problems [7,88,89].

**Bacteria and viruses**: which can cause infectious diseases such as hepatitis A, typhoid, and typhus [90,91].

**Industrial chemicals:** such as solvents, pesticides, and other chemicals that are harmful to human health and the environment [92,93].

**Persistent organic compounds (POPs):** such as dioxins and furans, which can cause cancer and other health problems [93,94]. For example, Rogers et al. (2021) describe a systematic approach to prioritizing landfill contaminants based on their toxicity. The approach uses a risk assessment model that considers factors such as acute and chronic toxicity, persistence, and bioaccumulation of contaminants.

The article highlights the importance of identifying and prioritizing landfill contaminants to inform decision making for leachate management and treatment. The authors suggest that this approach can be used by landfill operators, environmental agencies, and other stakeholders to identify contaminants of greatest concern and prioritize actions to reduce their impacts. The study also identifies some limitations and opportunities for future research in this area, such as the need to expand the landfill contaminant toxicity database, improve the accuracy of risk assessment models, and develop more effective strategies for managing landfill leachate. Overall, the article highlights the importance of a systematic approach to landfill leachate management and suggests that the proposed approach would be useful to improve environmental management and decision making for a bioremediation project for leachate-contaminated soils. The presence of these contaminants in leachate from landfills can have serious environmental and human health consequences. It is important to take measures to reduce and control the contamination of soil and groundwater by landfill leachate [95,96].

As discussed in this review, soil contamination by leachate is a major problem worldwide [85,96,97]. Leachate is a toxic liquid that seeps through waste in landfills and other waste disposal facilities and can contaminate nearby soil and groundwater [95]. The extent and severity of soil contamination by leachate vary by country and region. However, it is estimated that millions of tons of waste accumulate in landfills and other waste disposal facilities worldwide, which means that soil pollution by leachate is a widespread problem [98,99].

Table 6 provides a list of useful information for the scientific community, listing countries that have conducted research on the bioremediation of leachate-contaminated soils, mostly soils from open dumps or municipal soils; in some other cases, soils for agricultural structuring purposes, and in other cases, bioremediation of industrial soils. Developed countries have taken measures to reduce the amount of waste and to better manage the waste generated. However, as shown in Table 6, waste management in developing and underdeveloped countries is still inadequate, which increases the risk of soil contamination by leachate. In general, soil contamination by leachate is a major problem that requires global prevention and control measures. These include the use of safer and more sustainable waste disposal techniques and the development of waste management strategies that promote waste reduction, reuse, and recycling [98,99].

In this study, the prevalence of target contaminants for bioremediation was 53% for heavy metals. This is consistent with several authors [45,46,47,50,54,126,143], which are discussed in more detail in the following section. Although the type of heavy metals selected for bioremediation and the techniques used in each study vary, it is a fact that heavy metals are the pollutant of most concern worldwide. In second place, accounting for 26% of the cases of bioremediation of soils contaminated with leachate studied, are hydrocarbons. Although this is a widely studied issue, the cases of contamination by hydrocarbons presented here are limited to contamination by leachate, both in landfills and municipal soils, as shown in Table 6, the problem is broad and ranges from petroleum, methane, aromatic hydrocarbons, to pyrene, etc., each treated with different microorganisms, different techniques, and specific soils. Plastics or polymers are presented in 13% of the case studies in this systematic review. It is known that more than 300 million tons of plastics are produced in the world annually [144,145,146]. Currently, bioremediation of this type of pollutant is of great importance, since various microorganisms have been discovered that are able to biodegrade plastics of different classifications, such as polyethylene (PE), polyethylene terephthalate (PET), and polystyrene (PS), low-density polyethylene, as well as compounds used for its preservation. For example, dibutyl phthalate, also known as DBP, is an organic compound used in industry as a plasticizer, but also as an additive in adhesives, printing inks, and cosmetic products [80]. Another compound of this group, diethyl phthalate (DEP), is a colorless liquid with an unpleasant bitter taste. This synthetic compound is commonly used to add flexibility to plastics. It is used in products such as toothbrushes, car parts, tools, toys, and food packaging [29]. Finally, 8% of the case studies in this thesis are organic substances that are very diverse in nature and whose presence in soils is due to very different human activities such as agriculture, industry, transport, etc. The most common cause of universal contaminants is agriculture, since they are used to control parasites and plant diseases, to protect crops from harmful influences, even if they are not parasites, and to improve the quality and quantity of production, such as pesticides, herbicides, and fertilizers [147].

The studies listed in Table 6 show that bioremediation is an effective and sustainable option for treating contaminated soils. The results of these studies show that bioremediation is a promising technique for reducing contamination with heavy metals, hydrocarbons, polymers, and organic pollutants in soils.

#### 3.1.1. Heavy Metals

The studies reviewed indicate that bioremediation is effective in eliminating certain heavy metals, either individually or for specific groups. Forty-seven percent of the studies listed in Table 6 specifically address the bioremediation of heavy metals. In some of these studies, specific bacterial strains, mostly Pseudomonas, were isolated to treat soils contaminated with heavy metals [70,71,72,77,108,109,124,135] resulting in a high capacity of bacteria in bioremediation of mostly a specific contaminant. Compared to other studies using microbial consortia [11,12,13,14,15,16,101,103] where they showed promising results in the removal of a specific pollutant group or pollutant, in some cases, even symbiosis studies were performed with indigenous soil bacteria. Other studies investigated bioremediation using the accumulation and tolerance capacity of plants for the bioremediation of heavy metals [13,104,106], and the results indicated that it is a technique that can achieve high accumulation and tolerance capacity for this type of pollutant.

In other studies, such as Sedlakova-Kadukova et al. (2019), it was found that the combination of different bioremediation approaches, such as phytoextraction and rhizoremediation, was effective in reducing the levels of various heavy metals. However, some studies suggest that bioremediation may have limitations in terms of efficacy and treatment duration. For example, Elbehiry et al. (2020) found that bioremediation using bacteria was effective in reducing cadmium concentrations in contaminated soils, but the process was relatively slow and required a longer time to achieve significant reductions.

#### 3.1.2. Polymers and Their Derivatives

Nineteen percent of the studies listed in Table 6 deal with the bioremediation of soils contaminated with plastics, polymers, and their derivatives. This topic is of great environmental interest due to the amount of plastic waste generated today. The studies presented here demonstrate the effectiveness of various bioremediation techniques to degrade these contaminants in soils. Studies have found that the application of isolated bacterial strains along with the addition of nutrients significantly increases the degradation of these types of pollutants [100]. Moreover, studies in which a specific bacterium was used to degrade these types of pollutants [19,24,25,26,27,28] showed efficient bioremediation for a specific pollutant. Other studies showed that the use of a bacterial consortium increased the biodegradation potential [29,30,31,32], with a significant decrease in the amount of plastic after a 60-day treatment. Some other studies investigated the potential of an isolated bacterium and obtained a significant reduction in these compounds present in the soil [113]. An important result is the study conducted by Bardají et al. (2019), in which they demonstrated that oil-degrading bacteria have the potential to degrade polymers as well.

#### 3.1.3. Hydrocarbons

Eighteen percent of the studies listed in Table 6 address the bioremediation of soils contaminated with hydrocarbons, and although they use different strategies, all conclude that bioremediation is a viable and effective option for removing this type of contaminant. Most of these studies investigated the degradation capacity of this type of contaminant using bacterial consortia [17,18,19,20,21,22,23], which produced effective results in the removal of aromatic hydrocarbons and volatile organic compounds. In other studies, the degradation capacity of specific bacterial strains was evaluated by isolation and inoculation [132], showing a pollutant degradation capacity of up to 81% after 60 days [111]. Other studies have shown that phytoremediation in combination with bacterial strains could be a viable and effective alternative for the removal of hydrocarbons with removal up to 86.4% after 90 days, with plants and bacteria specific to the type of hydrocarbon to be removed [19].

#### 3.1.4. Organic Matter

The studies described below address the bioremediation of soils contaminated with organic pollutants using a variety of microorganisms and techniques, accounting for 16% of the case studies in Table 6. Most of these studies focus on evaluating the efficiency of bioremediation by bacterial consortia [38,39,40,41]. Some of these studies used *pseudomonas* [35,36,37] and other studies selected specific bacteria [33,34], which generally resulted in a high degradation capacity of these contaminants.

Overall, the studies evaluated indicate that bioremediation is a promising and effective technique for reducing the concentration of heavy metals and hydrocarbons and for biodegrading polymers and organic pollutants. However, the results depend on the type of pollutant and the specific soil conditions, and the effectiveness of bioremediation may depend on the specific treatment strategy and the combination of different bioremediation approaches.

### 3.2. Bioremediation in Different Countries around the World

As shown in Figure 4, China is the country with the most bioremediation studies on contaminated soils [105,107], followed by India [11,25,104,137,142], and third is Malaysia [30,70]. All these studies were conducted in most cases in industrial or municipal landfills or simply in synthetically contaminated soils to investigate bacterial behavior and biodegradation as a function of the specific contaminant selected, with heavy metals being the most commonly studied contaminants.

### 3.3. Impact of Heavy Metals

Heavy metals are toxic pollutants that can have significant adverse effects on the environment and human health. In the case of soils contaminated by leachates containing heavy metals, they can have negative effects on soil quality, microbial activity, and the health of surrounding ecosystems [91,138,148]. When heavy metals enter the soil via leachates, they can accumulate at the soil surface or penetrate into deeper layers, resulting in reduced soil fertility and microbial activity. Heavy metals can also accumulate in plants, which can affect their growth and nutrient quality, and they can be transferred through the food chain [15,113,149,150]. In humans, exposure to heavy metals can lead to a number of health problems, such as respiratory, cardiovascular, renal, and neurological diseases. In addition, heavy metal exposure can also be carcinogenic and mutagenic [95,143,151,152]. Enzyme activity in soils contaminated with heavy metals is significantly reduced (10–50 times) compared to non-contaminated soils [152]. Some examples of common heavy metals in this systematic review are Al, Co, Pb, As, Cr, Ni, Zn, Cd, and Cu [44,45,47,122,126,142]. Heavy metals remain in the soil for a considerable period of time [54,114,136,149]. Therefore, it is important to take measures to prevent the contamination of soils by leachates containing heavy metals and to remediate them when necessary. This may include the use of bioremediation strategies, phytostabilization, and other techniques to reduce soil toxicity and improve soil quality and microbial activity. In addition, it is important to continuously monitor soils near landfills and other waste sites to mitigate and prevent contamination by heavy metals and other toxic pollutants [47,102,153].

Figure 5 shows that heavy metals are among the pollutants of greatest concern worldwide and are the most studied pollutants, accounting for 53% of the total cases in this systematic review. The second most studied pollutants are hydrocarbons, which were included in this review because the majority of them deal with landfill soils (26% of all cases in this review). The third most studied pollutants in this report are organic pollutants with 13% of the case studies. Recently, some bacteria capable of biodegrading plastics such as polyethylene, polyethylene terephthalate, polystyrene, and others were discovered (8% of case studies) and offer promising applications for the future.

## 4. Conclusions

The concentrations and types of contaminants in soils around the world vary considerably. This is also true for the type of soil; although, they are mainly concentrated in landfills for all types of wastes, such as industrial, municipal, or urban. Different contamination rates have been analyzed for heavy metals, polymers, plastics, and organic pollutants. This systematic review contains results that depend on the type of bacteria or microorganisms used and the contaminant to be bioremediated, so it is subjective to ensure that a single bacterial species can bioremediate all contaminants presented here. However, this review provides a general overview of the types of bacteria that might be useful for a particular type of contaminant. It is necessary that soil contamination assessment and research be regulated worldwide, because it depends on the bioremediation needs of each site how the necessary components are used to achieve the expected result. Here, we have reviewed and presented a series of tables with all countries known to date to be subject to contamination, according to the PRISMA of a systematic review that has been used. The distribution of contamination worldwide is effectively demonstrated by the tables shown here. Different bioremediation strategies have been applied to reduce the negative impact on the soil and to preserve the ecological properties of these contaminated soils. This systematic review addresses bioremediation, specifically studies using microorganisms. Strategies to remediate toxic soil contaminants include chemical, biological, physical, electrical, and/or thermal mechanisms. The figures and tables in this systematic review summarize the importance of bioremediation in the world and the percentage of studies conducted in each country for the benefit of the soils in question. Soil bioremediation by isolation and inoculation of bacteria can be performed either in situ or ex situ; although, in most cases, it has been performed in controlled environments such as the laboratory. In some studies, bioaugmentation is recommended to achieve a higher percentage removal of the selected contaminant, especially in agricultural soils, as even 100% removal is possible and fertile soils can be achieved. The application of specific soil treatment techniques for each project depends on a variety of variables, such as the site and its contaminants, removal, mineralization, or elimination goals, cost-effectiveness, time, and cost. The variety of treatment methods helps in selecting the right bioremediation methods to use before conducting a large-scale remediation. The results summarized in this systematic review can help to develop innovative and applicable methods for assessing global soil contamination. In addition, these results can contribute to the development of innovative, applicable, and economically viable methods for the sustainable management of contaminated soils, whether from landfill leachate or other soil types, to reduce or eliminate risks to the environment and human health and to create a greener and more functional habitat on the planet.

As for future prospects, bioremediation remains an important technique for the remediation of contaminated soils. It is expected that new technologies will be developed and existing techniques will be improved to increase the effectiveness of bioremediation. One of the most important bioremediation techniques is biodegradation, which involves the use of microorganisms to degrade contaminants. To improve the results of biodegradation, techniques such as bioaugmentation (addition of microorganisms specifically selected to degrade contaminants) or biostimulation (addition of nutrients to increase microbial activity) can be used. Another important technique is phytoremediation, in which plants are used to absorb and degrade contaminants in the soil. In the future, it is expected that even more effective plants will be developed for phytoremediation and used in combination with biodegradation techniques.

In addition, the use of soil amendments can also improve the results of bioremediation. Soil amendments can improve soil conditions so that microorganisms can break down contaminants more effectively. In summary, bioremediation will continue to be an important technique for remediating contaminated soils in the future. Improved techniques of biodegradation, phytodiet, and the use of soil amendments will contribute to better bioremediation results in the future. The use of bacterial consortia could be an effective strategy for the bioremediation of soils contaminated by leachate, provided that the appropriate bacterial species are selected and environmental conditions are optimized to promote their growth and degradation activity.

## Figures and Tables

**Figure 1 microorganisms-11-00857-f001:**
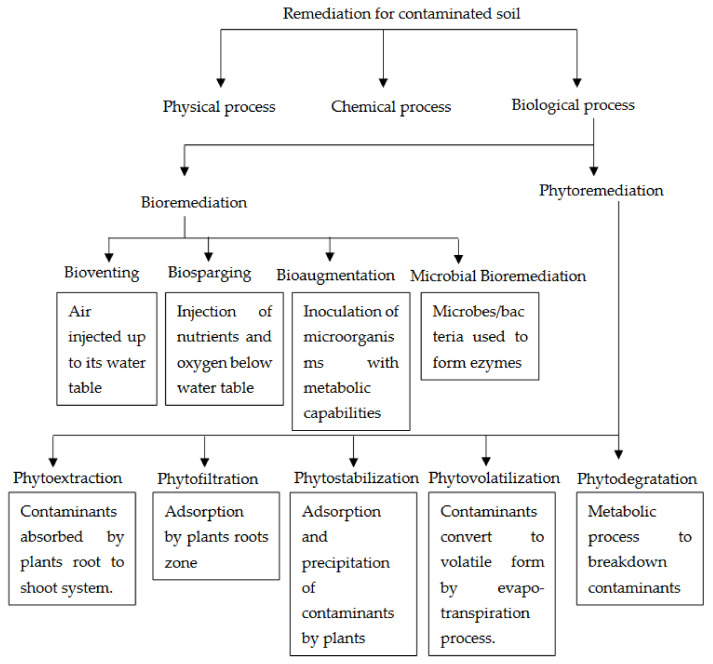
Remediation techniques [45].

**Figure 2 microorganisms-11-00857-f002:**
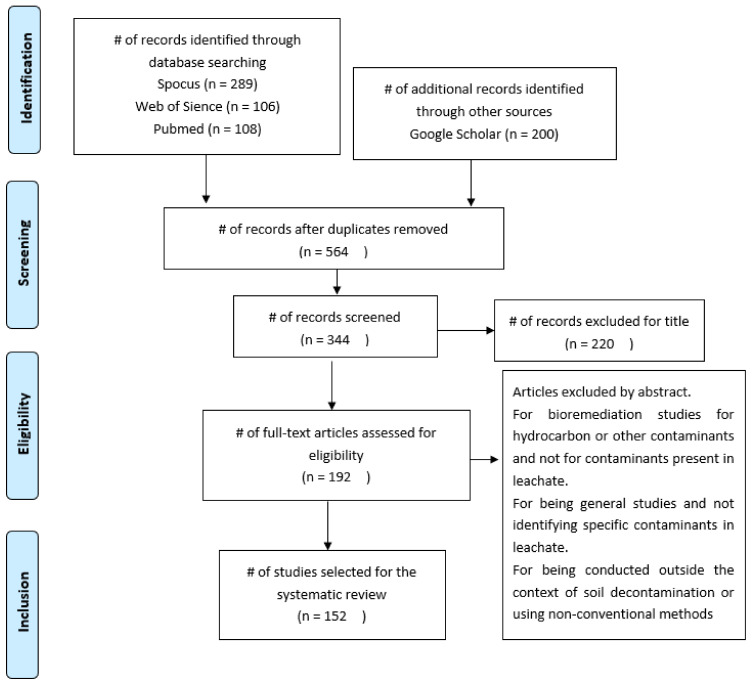
A diagram of the selected items, n is the number of articles. The complete list is shown in Appendix A.

**Figure 3 microorganisms-11-00857-f003:**
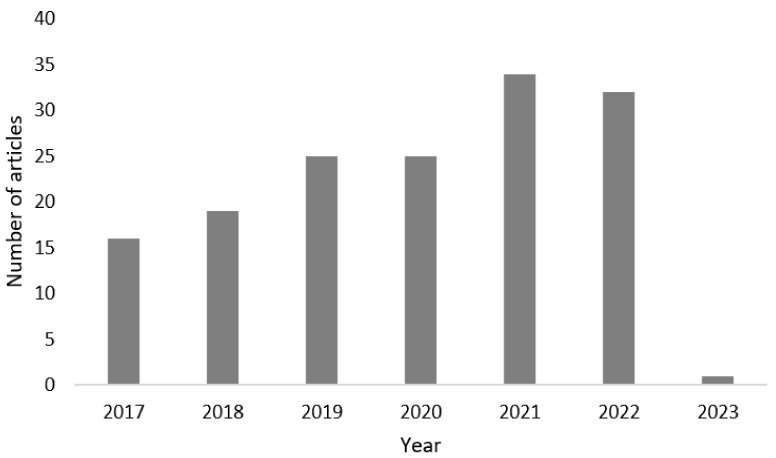
Articles published in the last 5 years (January 2017 to January 2023) on bioremediation of leachate-contaminated soils by treatment with microorganisms.

**Figure 4 microorganisms-11-00857-f004:**
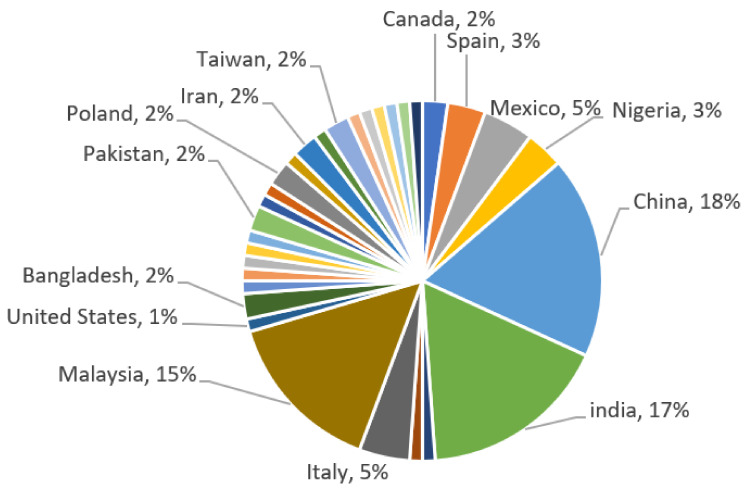
Percentage of research studies conducted by country of study.

**Figure 5 microorganisms-11-00857-f005:**
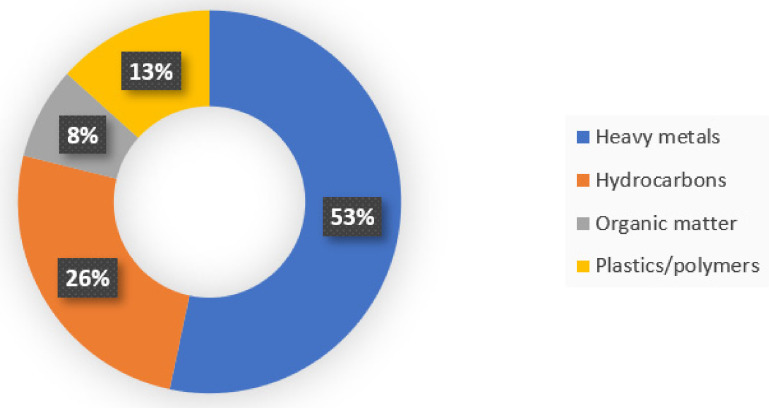
Types of pollutants studied for biological soil remediation.

**Table 1 microorganisms-11-00857-t001:** Search strategy with Boolean operators.

Database	Combination of Terms and Boolean Operators	Articles Found
Scopus	(TITLE-ABS-KEY (soil AND bioremediation AND leachate) OR TITLE-ABS-KEY (soil AND bioremediation AND landfill) OR TITLE-ABS-KEY (soil AND bioremediation AND techniques AND leachate AND landfill) AND NOT TITLE-ABS-KEY (hydrocarbons)) AND PUBYEAR > 2016 AND PUBYEAR > 2016	289
Web of Science	soil bioremediation AND leachate (All Fields) or soil bioremediation AND landfill (All Fields) or soil bioremediation techniques AND leachate AND landfill (All Fields) not Hydrocarbons	106
PubMed	(soil bioremediation AND leachate) OR (soil bioremediation AND landfill) OR (soil bioremediation techniques AND leachate AND landfill) NOT (hydrocarbons)	108
Google Scholar	“soil bioremediation” + “leachate” OR “soil bioremediation” + “landfill” OR “soil bioremediation techniques” + “leachate” + “landfill” − “hydrocarbons”	200

**Table 6 microorganisms-11-00857-t006:** Different pollutants studied in different countries with different soil types around the world.

Year	Country	Soil Type/Soil Use	Contaminant	Reference
2017	Malaysia	Landfill soil	NH(3)-N	[40]
2017	Malaysia	Landfill soil	Al, Cu, Cd, Mn, and Pb	[71]
2017	China	Agricultural soil	Di-(2-ethylhexyl) phthalate (DEHP)	[100]
2017	Malaysia	Landfill soil	Al, Cd, Cr, Fe, Ni, Pb, and Zn	[72]
2017	Spain	Not specified	Cu, Zn, Ni, Pb, and Cd	[101]
2017	Malaysia	Landfill soil	Pb, Al, and Mn	[70]
2017	Malaysia	Municipal soil	Pb, Al, and Cu	[102]
2017	Bulgaria	Landfill soil	Methane	[41]
2017	Malaysia	Landfill soil	Pb, As, Mn, Ni, and Cr	[103]
2017	India	Municipal soil	Cadmium and chromium	[104]
2018	Sri Lanka	Landfill soil	Fe, Mn, Cu, Ni, Cd, Zn, and Pb	[16]
2018	UK	Landfill soil	As and Hg	[105]
2018	Italy	Agricultural soil	trace elements and organic compounds	[34]
2018	India	Landfill soil	Fe, Mn, Zn, and Cu	[106]
2018	India	Landfill soil	Polyhydroxybutyrate	[24]
2018	India	Landfill soil	Nickel, cadmium, and chromium	[11]
2018	China	Landfill	Calcium chloride and urea	[107]
2018	Philippines	Landfill soil	Silver	[12]
2018	India	Not specified	Selenium	[13]
2018	Pakistan	Industrial soil	Chromium and arsenic	[108]
2018	Malaysia	Landfill soil	Mg, Al, Si, Cr, Mn, Cu, Ni, Co, Zn, As, Ag, Cd, Hg, and Pb	[109]
2018	Ireland	Landfill soil	Ammonia	[110]
2018	China	Landfill soil	Petroleum	[17]
2018	Romania	Industrial soil	Petroleum hydrocarbons	[111]
2018	Italy	Natural soil	Hydrocarbon	[112]
2019	China	Landfill soil	Cr, Ni, Pb, Mn, Cu, Zn, and Cd	[15]
2019	Mexico	Natural soil	polyacrylic and polyester polyurethane	[113]
2019	United States	Natural soil	*Escherichia coli* and *Rhodococcus erythropolis*	[114]
2019	China	Landfill soil	Diethyl phthalate (DEP)	[29]
2019	Slovakia	Not specified	Zn	[115]
2019	Switzerland	Landfill soil	As, Mn, Cu, Cr, and Fe	[74]
2019	India	Industrial soil (textile)	Color removal	[116]
2019	China	Industrial soil	Polycyclic aromatic hydrocarbons	[18]
2019	Brazil	Landfill soil	Polyethylene	[79]
2019	India	Landfill soil	Pyrene	[117]
2019	China	Not specified	Per- and polyfluorinated alkyl substances (PFAS) and lead and antimony	[118]
2019	China	Landfill soil	Ammonia	[33]
2019	Nigeria	Landfill soil	Cu and Pb	[119]
2019	China	Landfill soil	1-naphthol	[120]
2019	Australia	Landfill soil	Polycyclicaromatic hydrocarbon	[19]
2020	Canada	Landfill soil	Na and Cl	[84]
2020	China	Landfill soil	Particulate organic matter (POM)	[37]
2020	China	Landfill	H(2)S, NH(3), and VOC	[83]
2020	China	Natural soil	Pb, Cr, Cd, Cu, Mn, and Zn	[87]
2020	Morocco	Not specified	Organic pollutants	[39]
2020	Malaysia	Landfill soil	Fe, Cu, and Cr	[76]
2020	India	Industrial soil	Petroleum hydrocarbons	[20]
2020	Malaysia	Municipal soil	Di-(2-ethylhexyl)	[30]
2020	Poland	Cultivated soil	Polylactide (PLA) and polyethylene terephthalate (PET)	[31]
2020	Bangladesh	Landfill soil	Chromium (VI)	[75]
2020	Mexico	Mining soil	As, Pb, Cu, Mn, and Fe	[121]
2020	Egypt	Landfill soil	Cr, Cu, Fer, Mn, Ni, Pb, and Zn	[122]
2020	Iran	Landfill soil	Copper	[123]
2020	China	Natural soil	Polyethylene (PE), polystyrene (PS), polypropylene (PP), polyvinyl chloride (PVC), polyurethane (PUR), and polyethylene terephthalate (PET)	[32]
2020	China	Landfill soil	Volatile organic compounds (VOCs)	[35]
2020	India	Landfill soil	Pyrene	[81]
2020	Malaysia	Landfill soil	Ni, Pb, and Zn	[77]
2020	Malaysia	Landfill soil	Pyrene and cadmium	[124]
2020	Nigeria	Industrial soil	Polycyclic aromatic hydrocarbons (PAHs), pesticides, petroleum products, volatile organic compounds (VOCs), organic solvents, heavy metals (not specified)	[21]
2020	Poland	Mixed soil	Polychlorinated biphenyls (PCBs) and petroleum hydrocarbons (TPH)	[22]
2021	Spain	Landfill soil	Cd; Ni, Pb; Cr and benzo(a)pyrene	[125]
2021	Nigeria	Landfill soil	Cd and Pb	[126]
2021	India	Landfill soil	polyethylene (PE), polyethylene terephthalate (PET), and polystyrene (PS)	[25]
2021	Bangladesh	Industrial soil	Diesel	[23]
2021	Indonesia	Landfill soil	Hg, Cd, Pb, Mg, Zn, Fe, Mn, and Cu	[127]
2021	Slovakia	Artificial soil	Pb, Zn, Cu, or Ni	[128]
2021	India	Municipal solid	Methane	[36]
2021	Iran	Landfill soil	Low-density polyethylene	[26]
2021	Peru	Agricultural soil	Pb	[129]
2021	India	Municipal soil	Decabrominated diphenyl ether	[27]
2021	Taiwan	See soil	Cd	[130]
2021	Italy	Landfill soil	Low-density polyethylene	[131]
2021	Taiwan	Not specified	Petroleum hydrocarbon	[132]
2021	China	Natural soil	Total petroleum hydrocarbons (TPH)	[133]
2021	China	Not specified	Di-(2-ethylhexyl) phthalate (DEHP)	[134]
2022	Mexico	Landfill soil	Phenolic compounds and phthalates	[28]
2022	Spain	Garden soil	Pb	[78]
2022	India	Municipal soil	Cd	[14]
2022	Germany	Landfill soil	Dibutyl phthalate	[80]
2022	Malaysia	Landfill soil	As, Cr, Cu, Fe, Mn, Ni, Pb, and Zn	[135]
2022	Pakistan	Natural soil	Pb, Cu, Zn, Fe, and Cr	[136]
2022	Korea	Landfill soil	Oil and hydrocarbons	[82]
2022	India	Landfill soil	Low-density polyethylene (LDPE)	[137]
2022	Mexico	Mining soil	As	[138]
2022	Italy	Landfill soil	Polyethylene, polyvinyl chloride, and polyethylene terephthalate	[139]
2022	Malaysia	Landfill soil	Pb, Cu, As, Mn, Cr, Zn, Fe, and Ni	[140]
2022	Canada	Agricultural soil	Pesticide	[38]
2022	China	Natural soil	Polycyclic aromatic hydrocarbon (PAH)	[141]
2023	India	Landfill soil	Cd, Pb, Ni, and Cr	[142]

## Data Availability

The data presented in this study are available on request from the corresponding author.

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
