# Peer review of "Global Situation of Bioremediation of Leachate-Contaminated Soils by Treatment with Microorganisms: A Systematic Review"

_microorganisms, 2023, doi:10.3390/microorganisms11040857_

Round 1

Reviewer 1 Report

Abstract, Line 11: “The aim of this systematic review is to present…”.

I suggest you write “This systematic review presents…”.

Lines 29-30: “open landfills”.

I suggest you write “open landfills and dumpsites”. Indeed, uncontrolled sites where waste is disposed of are defined as “dumpsites”. As a general rule, such sites don’t have any geomembrane or waterproof layer at their bottom; thus, leachate can easily contaminate the soil and the groundwater. They are very common in Low- and Lower-Middle income countries. If needed, you can find a lot of information about it in the scientific literature (for instance, you can write “dumpsites developing countries” in some search engines).

Lines 42-43: “These environmentally friendly remediation processes are referred to as bioremediation”.

I would reformulate the sentence. Indeed, it would be more appropriate to write that these biological treatments usually have a very low environmental impact. For example, Bioventing and Biosparging (which you included in Figure 1) cannot be considered totally environmentally friendly in all cases.

General comments about the Introduction section.

You should expand it. First of all, previous reviews and research on the topic you mentioned should be discussed. For instance, you cited ref. [9] (i.e. “Review on bioremediation and phytoremediation techniques of heavy metals in contaminated soil from dump site”), but you should discuss this and other reviews highlighting the difference with your review. For instance, the authors of this recent publication assessed something very similar to your review, i.e. contamination of soil due to dumpsites. As I mentioned, it can be traced back to the waste disposal sites under your discussion.

Furthermore, the novelty (if any) presented by your review compared to other studies should be discussed.

Lines 97-99: “Inoculation of nutrients such as nitrogen or phosphorus along with oxygen is the process behind bioventing.”

The sentence is inaccurate. Inoculation of nutrients is not always required.

Section 1.1.2. Biosparging technique.

Biosparging should be discussed with more accuracy.

Section 1.1.3. Bioaugmentation technique

Please, provide more details about bioaugmentation.

Lines 151-152: “PRISM guidelines”.

Please, correct it by writing “PRISMA guidelines”.

Line 156:published between 2017 and 2023”.

Please, recheck the sentence. Are you sure you searched for articles published until 2023? Indeed, we are only at the beginning of February 2023.

Furthermore, if possible, it would be helpful to specify the month (for example, “from January 2017 to…”).

Figure 2.

It would be useful to provide, at least, the list of the 152 studies that you selected for the systematic review.  For example, you can provide it as Supplementary Material.

Table 2.

Studies should be better organized in the table. Currently, they seem to be placed randomly without any order. Furthermore, some words need to be rechecked (for instance, you wrote “contaminante”).

Lines 281-282: “the most commonly used method for bioremediation is isolation and inoculation of various bacteria”.

Please, explain how inoculation usually takes place.

Lines 290-291: “It has been found that the main problem with this type of soil is heavy metal contamination”.

The sentence is a bit inaccurate. For instance, Rogers et al. (2021) have recently presented a list of contaminants that can be found in landfill leachate. The authors ranked such pollutants by their potential toxicity. Among the most dangerous, many organic pollutants were mentioned. Thus, I suggest you significantly improve your study, giving more importance to them. For some clues, you can have a look at Table 1 from Rogers et al. (2021) (here: Rogers, E. R., Zalesny, R. S., Jr, & Lin, C. H. (2021). A systematic approach for prioritizing landfill pollutants based on toxicity: Applications and opportunities. Journal of environmental management, 284, 112031. https://doi.org/10.1016/j.jenvman.2021.112031).

Section 3. Results and Discussion.

Table 3 has a lot of useful information. However, you should discuss it in detail. Currently, you said almost nothing about it. Thus, your review appears valuable for the scientific community until the Materials and Methods section.

Line 324: “76, 118, 119”.

Maybe, you forgot the brackets.

Line 325: “Metals that occur naturally in the soil do not cause contamination”.

The sentence seems inaccurate. For instance, Arsenic occurs naturally in the soil and can cause a lot of health problems for people drinking water contaminated by it.

Line 355: “prism”.

The correct word is PRISMA.

Section 4. Conclusions.

Perspectives for future research should be discussed.

Additional considerations.

As I mentioned (see comment about lines 290-291), in your review, and in particular in your discussion, you should give more importance to organic pollutants as well.

Additionally, the results of table 3 must be discussed in detail. Removal percentages, strengths and weaknesses of the studies you found need to be highlighted as well. The discussion section is currently very weak.

The bioremediation strategies you found to remove such pollutants need to be discussed in detail.

Author Response

Dr. Martin von Bergen

Microorganisms

Editor-in-Chief

March 5th, 2023 

Concerning the work Ms. No. microorganisms-2231004 entitled “Global Situation of Bioremediation of Leachate-Contaminated Soils by Treatment with Microorganisms: A Systematic Review” that we have submitted for consideration to be published in Microorganisms, here we present the comments related to the points raised by the reviewers that kindly reviewed the first version of the manuscript.

Reviewer #1:

1) Abstract, Line 11: “The aim of this systematic review is to present…”.

I suggest you write “This systematic review presents…”

Dear Reviewer, thank you for your suggestion. The text has been changed, as indicated in the "abstract section".

2) Lines 29-30: “open landfills”.

I suggest you write “open landfills and dumpsites”. Indeed, uncontrolled sites where waste is disposed of are defined as “dumpsites”. As a general rule, such sites don’t have any geomembrane or waterproof layer at their bottom; thus, leachate can easily contaminate the soil and the groundwater. They are very common in Low- and Lower-Middle income countries. If needed, you can find a lot of information about it in the scientific literature (for instance, you can write “dumpsites developing countries” in some search engines).

We agree with your observation and recommendation. Changes were made to the suggested words and further research was "open landfill and dumpsites" and “dumpsites developing countries”, as you suggested. This information has been included in the "Introduction section" in the manuscript.

3) Lines 42-43: “These environmentally friendly remediation processes are referred to as bioremediation”.

I would reformulate the sentence. Indeed, it would be more appropriate to write that these biological treatments usually have a very low environmental impact. For example, Bioventing and Biosparging (which you included in Figure 1) cannot be considered totally environmentally friendly in all cases.

Dear Reviewer thank you for your suggestions to enrich our work, now the sentence has been reformulated, correcting the above mentioned.

4) General comments about the Introduction section.

You should expand it. First of all, previous reviews and research on the topic you mentioned should be discussed. For instance, you cited ref. [9] (i.e. “Review on bioremediation and phytoremediation techniques of heavy metals in contaminated soil from dump site”), but you should discuss this and other reviews highlighting the difference with your review. For instance, the authors of this recent publication assessed something very similar to your review, i.e. contamination of soil due to dumpsites. As I mentioned, it can be traced back to the waste disposal sites under your discussion.

Furthermore, the novelty (if any) presented by your review compared to other studies should be discussed.

Thank you for your suggestion. We have expanded the Introduction section, section 1.1, section 1.2, emphasizing research prior to this systematic review and comparing some of the techniques used. We have also highlighted the purpose of this review and the usefulness in comparison with other studies.

5) Lines 97-99: “Inoculation of nutrients such as nitrogen or phosphorus along with oxygen is the process behind bioventing.” 

The sentence is inaccurate. Inoculation of nutrients is not always required.

We agree with your suggestion. The paragraph has been rewritten, correcting the error you mention.

6). Section 1.1.2. Biosparging technique.

Biosparging should be discussed with more accuracy.

Dear reviewer, thank you for your suggestions. The definition has been changed and explained with more accuracy in section 1.1.2 which is now section 1.2.2.

7) Section 1.1.3. Bioaugmentation technique

Please, provide more details about bioaugmentation.

The paragraph of section 1.1.3 has been rewritten, which is now section 1.2.3, detailing and clearly defining this technique, and adding the section "1.3. Phytoremediation technology" due to the fact that the introduction section mentions the use of the phytoremediation technique in previous works to this systematic review.

8) Lines 151-152: “PRISM guidelines”.

Please, correct it by writing “PRISMA guidelines”.

Dear Reviewer this was a mistake that has been now corrected in the "Materials and Methods section".

9) Line 156: “published between 2017 and 2023”.

Please, recheck the sentence. Are you sure you searched for articles published until 2023? Indeed, we are only at the beginning of February 2023.

Furthermore, if possible, it would be helpful to specify the month (for example, “from January 2017 to…”).

We agree and aprecciate your suggestion. In fact, the systematic review is limited from January 2017 to January 2023. This has now been corrected in the “Initial Search section” and Figure 3 description.

10) Figure 2.

It would be useful to provide, at least, the list of the 152 studies that you selected for the systematic review.  For example, you can provide it as Supplementary Material.

We agree with your observation. Therefore, the information on supplementary material is mentioned in the description in Figure 2. The list of 152 studies is attached like Suplementary material S1.

11) Table 2.

Studies should be better organized in the table. Currently, they seem to be placed randomly without any order. Furthermore, some words need to be rechecked (for instance, you wrote “contaminante”).

Dear reviewer. We agree with your observation and recommendation. In this sense, Table 2 was grouped according to the type of  pollutant to bioremediate. Taking into account the year of publication, from the old to the current year. This information has been included in the “Results and Discussion section”.

12) Lines 281-282: “the most commonly used method for bioremediation is isolation and inoculation of various bacteria”.

Please, explain how inoculation usually takes place.

Thank you for your suggestion. The results and discussion section of the manuscript has been modified, explaining the incubation, inoculation and isolation of bacteria. As well as highlighting the success cases in Table 2, Table 3, Table 4, and Table 5 on the same section.

13) Lines 290-291: “It has been found that the main problem with this type of soil is heavy metal contamination”.

The sentence is a bit inaccurate. For instance, Rogers et al. (2021) have recently presented a list of contaminants that can be found in landfill leachate. The authors ranked such pollutants by their potential toxicity. Among the most dangerous, many organic pollutants were mentioned. Thus, I suggest you significantly improve your study, giving more importance to them. For some clues, you can have a look at Table 1 from Rogers et al. (2021) (here: Rogers, E. R., Zalesny, R. S., Jr, & Lin, C. H. (2021). A systematic approach for prioritizing landfill pollutants based on toxicity: Applications and opportunities. Journal of environmental management, 284, 112031. https://doi.org/10.1016/j.jenvman.2021.112031).

Thank you for your suggestion. Section 3.1, “The contamination of leachate-contaminated soils worldwide” in the manuscript has been modified to emphasize the main contaminants that are present in leachate by analyzing the reference suggested by the Reviewer have now been included in the text.

14) Section 3. Results and Discussion.

Table 3 has a lot of useful information. However, you should discuss it in detail. Currently, you said almost nothing about it. Thus, your review appears valuable for the scientific community until the Materials and Methods section.

Attending the Reviewer's observation. We have carried out a comprehensive revision of the data in section 3 "Results and Discussion". We have arranged Table 3, which is now Table 6, of the investigations in chronological order and written reviews of some of the investigations to highlight the importance of this systematic review.

15) Line 324: “76, 118, 119”.

Maybe, you forgot the brackets.

Dear reviewer, it was a mistake, which has been corrected in the “impact of heavy metals section”.

16) Line 325: “Metals that occur naturally in the soil do not cause contamination”.

The sentence seems inaccurate. For instance, Arsenic occurs naturally in the soil and can cause a lot of health problems for people drinking water contaminated by it.

The authors appreciate the Reviewer suggestions to enrich the work. Attending your observation, the "Impact of heavy metals" section has been now re-written including more recent references and to highlight the importance of soil contamination by heavy metals.

17) Line 355: “prism”.

The correct word is PRISMA.

Dear Reviewer this was a mistake that has been now corrected in the Conclusion section.

18) Section 4. Conclusions.

Perspectives for future research should be discussed.

 Dear Reviewer thank you for your suggestions to enrich our work. The conclusion section, now includes discussion of the future prospects of this systematic review.

19) Additional considerations.

As I mentioned (see comment about lines 290-291), in your review, and in particular in your discussion, you should give more importance to organic pollutants as well.

Additionally, the results of table 3 must be discussed in detail. Removal percentages, strengths and weaknesses of the studies you found need to be highlighted as well. The discussion section is currently very weak.

The bioremediation strategies you found to remove such pollutants need to be discussed in detail.

We agree with your observation and recommendation. In this sense, radical changes were made in the discussion section of results and conclusions as well as analysis of the studies addressed in the review. We also rearranged the tables and discussed some of the studies contained therein. Achieving a change and highlighting the importance of this systematic review.

Dear Reviewers, thank you very much in advance for your kind consideration to our work.

Sincerely yours,

Dr. José Alberto Rodríguez Morales

Facultad de Ingeniería

Campus UAQ-Aeropuerto

Carr. A Chichimequillas S/N, Terrenos Ejidales Bolaños, Querétaro, Qro. Cp. 76140

Universidad Autónoma de Querétaro

Tel. 442.237.69.16

http://www.uaq.mx/ingenieria/

http://aeropuerto.uaq.mx/

Reviewer 2 Report

The manuscript entitled ‘Global Situation of Bioremediation of Leachate-Contaminated Soils by Treatment with Microorganisms: A Systematic Review’ summarized the current trends of bioremediation of leachate-contaminated soils, which is a good point. However, the manuscript is not well organized, and some key issues are missing.

1.    Line 108-111. Bioaugmentation is not only suitable for soils contaminated by chlorine, many other pollutants are also feasible.

2.    Table 2 and table 3 can be organized in a more logical way for the readers to have a better view of the current works. Maybe similar contaminants, similar microorganisms or similar soil types can be clustered.

3.    More comprehensive analysis is needed for table 2 and table 3. Microorganisms are very important for bioremediation, so the authors should analyze the roles of the microorganisms for bioremediation, their enzymes, and the kind of contaminants they can be applied in bioremediation. Also, bacteria and fungi are to a great extent different in bioremediation, this point should be discussed as well.

4.    Bacterial consortium usually is more powerful in bioremediation than individual bacteria, therefore this needs specifical concern.

5.    Sometimes, bioremediation is not sufficient to realized satisfactory bioremediation, therefore other techniques can be combined with bioremediation. This point is also very important.

6.    Section 3.3: not only heavy metals, other organic pollutants, and emerging contaminants should also be discussed.

7.    Conclusion is not profound for a review paper, as the lack of current work and future prospective must be presented.

Author Response

Dr. Martin von Bergen

Microorganisms

Editor-in-Chief

 March 5th, 2023 

Concerning the work Ms. No. microorganisms-2231004 entitled “Global Situation of Bioremediation of Leachate-Contaminated Soils by Treatment with Microorganisms: A Systematic Review” that we have submitted for consideration to be published in Microorganisms, here we present the comments related to the points raised by the reviewers that kindly reviewed the first version of the manuscript.

Reviewer #2:

 The manuscript entitled ‘Global Situation of Bioremediation of Leachate-Contaminated Soils by Treatment with Microorganisms: A Systematic Review’ summarized the current trends of bioremediation of leachate-contaminated soils, which is a good point. However, the manuscript is not well organized, and some key issues are missing

1) Line 108-111. Bioaugmentation is not only suitable for soils contaminated by chlorine, many other pollutants are also feasible.

Dear Reviewer, thank you for your suggestion. We have rewritten the information in "section 1.2.3. Bioaugmentation technique", focusing on the scopes and limitations.

2) Table 2 and table 3 can be organized in a more logical way for the readers to have a better view of the current works. Maybe similar contaminants, similar microorganisms or similar soil types can be clustered.

We agree with your observation. Thus, Table 2 has been organized by types of contaminants, now they are Table 2, Table 3, Table 4, and Table 5. Table 3, which is now table 6, was organized by year of publication. In all the tables, an analysis of some of the studies contained therein was made in order to highlight the importance of this systematic review.

3) More comprehensive analysis is needed for table 2 and table 3. Microorganisms are very important for bioremediation, so the authors should analyze the roles of the microorganisms for bioremediation, their enzymes, and the kind of contaminants they can be applied in bioremediation. Also, bacteria and fungi are to a great extent different in bioremediation, this point should be discussed as well.

The authors appreciate the Reviewer suggestion. We have included discussion and contrast of some of the studies presented in table 2, which are now table 2 to table 5. Likewise, table 3, which is now table 6, has included discussion of the results of some of the studies presented in the tables. Emphasizing the importance of the variables to be taken into account to carry out bioremediation in soils contaminated by leachates.

4) Bacterial consortium usually is more powerful in bioremediation than individual bacteria, therefore this needs specifical concern.

Thank you for your suggestion. the "discussion of results and conclusions" section has been rewritten, in which the importance of using bacterial consortia and the importance of bacterial species selection have been mentioned. 

5) Sometimes, bioremediation is not sufficient to realized satisfactory bioremediation, therefore other techniques can be combined with bioremediation. This point is also very important.

The authors appreciate the Reviewer recommendation. In section 1.2.2, 1.2.3, 1.2.4 and 1.3, the main bioremediation techniques are highlighted and in the discussion of results and conclusion section the most commonly used techniques are highlighted. However, by adding information on the different remediation techniques and not on bioremediation, the focus of this work would be lost, since the PRISMA Systematic Review focused only on bioremediation techniques, excluding remediation techniques.

6). Section 3.3: not only heavy metals, other organic pollutants, and emerging contaminants should also be discussed.

Dear reviewer, we agree with your observation. A brief description of each of the important contaminants to bioremediate has been included in Section 3.1 "Contamination of leachate-contaminated soils worldwide". At several points in the manuscript, the types of contaminants in leachate-contaminated soils are mentioned, as well as discussion of some of the studies in this systematic review. It is argued that not only heavy metals are important for bioremediation. It is important to mention that section 3.3 has been rewritten. However, it focuses only on heavy metals because Figure 5 highlights that 53% of the studies analyzed in this systematic review focused on bioremediation of heavy metals.

7) Conclusion is not profound for a review paper, as the lack of current work and future prospective must be presented.

Dear Reviewer thank you for your suggestions to enrich our work. The conclusion section, now has been extended and includes discussion of the future prospects of this systematic review.

Dear Reviewers, thank you very much in advance for your kind consideration to our work.

Sincerely yours,

Dr. José Alberto Rodríguez Morales

Facultad de Ingeniería

Campus UAQ-Aeropuerto

Carr. A Chichimequillas S/N, Terrenos Ejidales Bolaños, Querétaro, Qro. Cp. 76140

Universidad Autónoma de Querétaro

Tel. 442.237.69.16

http://www.uaq.mx/ingenieria/

http://aeropuerto.uaq.mx/

Round 2

Reviewer 1 Report

The manuscript was improved, but it was not enough. Indeed, previous reviews on the topic should have been discussed more adequately. Furthermore, the novelty of the current study remains uncertain. Finally, some improvements were not adequate.

Lines 76-79: “Bioremediation technologies are based on redox processes that focus on altering the chemistry and microbiology of water by injecting selected reagents into contaminated water to enhance the degradation and extraction of various contaminants through in situ chemical oxidation/reduction reactions [12], [13].”

The definition appears incorrect and incomplete. In particular, “by injecting selected reagents into contaminated water”.

Lines 84-88: “In recent decades, various technologies have been developed to remediate soil and 84 groundwater contamination. However, this review focuses on biological treatments, 85 which include several contaminant removal techniques, also known as bioremediation 86 [14], [15]. The following is an analysis of the techniques used in studies conducted prior 87 to this systematic review.”

In my previous, I asked you to discuss previous reviews of the topic you addressed adequately. I also asked you to adequately highlight the novelty of your study. Unfortunately, it seems you have not adequately addressed my request.

Lines 137-138: “This technique is the most commonly used in situ mechanism in which air and nutrients are added to polluted soil to stimulate microorganisms.”

I previously asked you to correct the sentence. It seems your changes kept the sentence incorrect. Indeed, it is not always required to add nutrients in bioventing.

Lines 454-456: “The studies mentioned in Table 6 are cronological studies. These studies are aimed at bioremediation of leachate contaminated soils using different approaches and treatment strategies. The results of some of the studies are briefly discussed below”.

These lines should be improved. Furthermore, it is a bit unconventional the way of discussing the studies. It is not clear why you decided to discuss only some studies.

References.

Recheck them. Indeed, I noted some typos or lack of information (e.g. [7], [10], [11]).

Author Response

Dr. Martin von Bergen

Microorganisms

Editor-in-Chief

March 16th, 2023 

Concerning the work Ms. No. microorganisms-2231004 entitled “Global Situation of Bioremediation of Leachate-Contaminated Soils by Treatment with Microorganisms: A Systematic Review” that we have submitted for consideration to be published in Microorganisms, here we present the comments related to the points raised by the reviewers that kindly reviewed the second version of the manuscript.

Reviewer #1: The manuscript was improved, but it was not enough. Indeed, previous reviews on the topic should have been discussed more adequately. Furthermore, the novelty of the current study remains uncertain. Finally, some improvements were not adequate.

Lines 76-79: “Bioremediation technologies are based on redox processes that focus on altering the chemistry and microbiology of water by injecting selected reagents into contaminated water to enhance the degradation and extraction of various contaminants through in situ chemical oxidation/reduction reactions [12], [13].”

The definition appears incorrect and incomplete. In particular, “by injecting selected reagents into contaminated water”.

Dear Reviewer, it was a mistake that has been now corrected in 1.1 section.

 Lines 84-88: “In recent decades, various technologies have been developed to remediate soil and 84 groundwater contamination. However, this review focuses on biological treatments, 85 which include several contaminant removal techniques, also known as bioremediation 86 [14], [15]. The following is an analysis of the techniques used in studies conducted prior 87 to this systematic review.”

In my previous, I asked you to discuss previous reviews of the topic you addressed adequately. I also asked you to adequately highlight the novelty of your study. Unfortunately, it seems you have not adequately addressed my request.

We thank the reviewer for his suggestion and sincerely apologize for the error. The section on "Bioremediation Techniques" has been rewritten. The analysis of the studies presented in this systematic review is detailed in the "Results and Discussion" section.

Lines 137-138: “This technique is the most commonly used in situ mechanism in which air and nutrients are added to polluted soil to stimulate microorganisms.”

I previously asked you to correct the sentence. It seems your changes kept the sentence incorrect. Indeed, it is not always required to add nutrients in bioventing.

Dear Reviewer, Section 1.2.1 Biobenting technique has been rewritten.

 Lines 454-456: “The studies mentioned in Table 6 are cronological studies. These studies are aimed at bioremediation of leachate contaminated soils using different approaches and treatment strategies. The results of some of the studies are briefly discussed below”.

-These lines should be improved. Furthermore, it is a bit unconventional the way of discussing the studies. It is not clear why you decided to discuss only some studies.

 The authors appreciate the Reviewer suggestions to enrich the work. Attending your observation. The studies listed in Table 6 have now been adequately discussed. Adding section 3.1.1, 3.1.2, 3.1.3, and 3.1.4, in order to discuss the results according to the type of pollutant to bioremediate and finally mentioning future projections.References.

Recheck them. Indeed, I noted some typos or lack of information (e.g. [7], [10], [11]).

Attending the Reviewer´s observation, we have we have corrected the suggested references.

Dear Reviewers, thank you very much in advance for your kind consideration to our work.

Sincerely yours,

Dr. José Alberto Rodríguez Morales

Facultad de Ingeniería

Campus UAQ-Aeropuerto

Carr. A Chichimequillas S/N, Terrenos Ejidales Bolaños, Querétaro, Qro. Cp. 76140

Universidad Autónoma de Querétaro

Tel. 442.237.69.16

http://www.uaq.mx/ingenieria/

http://aeropuerto.uaq.mx/

Reviewer 2 Report

This manuscript can be accepted in the current form.

Author Response

Dr. Martin von Bergen

Microorganisms

Editor-in-Chief

March 16th, 2023 

Concerning the work Ms. No. microorganisms-2231004 entitled “Global Situation of Bioremediation of Leachate-Contaminated Soils by Treatment with Microorganisms: A Systematic Review” that we have submitted for consideration to be published in Microorganisms, here we present the comments related to the points raised by the reviewers that kindly reviewed the second version of the manuscript.

Reviewer #2: This manuscript can be accepted in the current form.

Dear Reviewer, thank you very much for your valuable comments and your time to improve the quality of our work

Sincerely yours,

Dr. José Alberto Rodríguez Morales

Facultad de Ingeniería

Campus UAQ-Aeropuerto

Carr. A Chichimequillas S/N, Terrenos Ejidales Bolaños, Querétaro, Qro. Cp. 76140

Universidad Autónoma de Querétaro

Tel. 442.237.69.16

http://www.uaq.mx/ingenieria/

http://aeropuerto.uaq.mx/
